# miRNAs Related to Immune Checkpoint Inhibitor Response: A Systematic Review

**DOI:** 10.3390/ijms25031737

**Published:** 2024-02-01

**Authors:** José Luis García-Giménez, Wiam Saadi, Angel L. Ortega, Agustin Lahoz, Guillermo Suay, Julián Carretero, Javier Pereda, Ahlam Fatmi, Federico V. Pallardó, Salvador Mena-Molla

**Affiliations:** 1Department of Physiology, Faculty of Medicine and Dentistry, University of Valencia, 46010 Valencia, Spain; j.luis.garcia@uv.es (J.L.G.-G.); federico.v.pallardo@uv.es (F.V.P.); 2INCLIVA Health Research Institute, INCLIVA, 46010 Valencia, Spain; 3Consortium Center for Biomedical Network Research on Rare Diseases (CIBERER), Institute of Health Carlos III, 46010 Valencia, Spain; 4Department of Biology, Faculty of Nature, Life and Earth Sciences, University of Djillali Bounaama, Khemis Miliana 44225, Algeria; w.saadi@univ-dbkm.dz; 5Department of Physiology, Faculty of Pharmacy, University of Valencia, 46100 Burjassot, Spain; angel.ortega@uv.es (A.L.O.); julian.carretero@uv.es (J.C.); javier.pereda@uv.es (J.P.); 6Biomarkers and Precision Medicine Unit, Health Research Institute-Hospital La Fe, 46026 Valencia, Spain; agustin.lahoz@uv.es; 7Analytical Unit, Health Research Institute-Hospital La Fe, 46026 Valencia, Spain; 8Medical Oncology Department, Hospital Universitari i Politècnic La Fe, 46026 Valencia, Spain; guillermo_suay@iislafe.es; 9Department of Microbiology & Biochemistry, Faculty of Science, University of M’sila, M’sila 28000, Algeria; ahlem.fatmi@gmail.com

**Keywords:** cancer immunity, epigenetic regulation, immune checkpoint inhibitor, microRNAs, systematic review

## Abstract

The advent of immune checkpoint inhibitors (ICIs) has represented a breakthrough in the treatment of many cancers, although a high number of patients fail to respond to ICIs, which is partially due to the ability of tumor cells to evade immune system surveillance. Non-coding microRNAs (miRNAs) have been shown to modulate the immune evasion of tumor cells, and there is thus growing interest in elucidating whether these miRNAs could be targetable or proposed as novel biomarkers for prognosis and treatment response to ICIs. We therefore performed an extensive literature analysis to evaluate the clinical utility of miRNAs with a confirmed direct relationship with treatment response to ICIs. As a result of this systematic review, we have stratified the miRNA landscape into (i) miRNAs whose levels directly modulate response to ICIs, (ii) miRNAs whose expression is modulated by ICIs, and (iii) miRNAs that directly elicit toxic effects or participate in immune-related adverse events (irAEs) caused by ICIs.

## 1. Introduction

Study of the tumor microenvironment has uncovered a wide battery of mechanisms exploited by neoplastic cells to evade immune system action, which enables them to survive and fosters tumorigenesis [1]. Immune cell activation requires T cell receptor (TCR) recognition of antigens presented by major histocompatibility complex class I or II molecules (MHC-I/II) expressed on normal cells or antigen presenting cells (APCs), and a costimulatory pathway wherein the receptor CD28 binds to ligands expressed on APC membrane such as B7 family ligands B7-1/CD80 or B7-2/CD86. This interaction stabilizes the signal and triggers the complete activation, proliferation, survival, and cytokine production of T lymphocytes. This same signaling initiates a regulatory mechanism to prevent overactivation of the immune system, in which the expression of immune checkpoint (IC) molecules such as cytotoxic T lymphocyte-associated antigen-4 (CTLA-4; CD152) is upregulated on the lymphocyte cell membrane after T cell activation to compete for the same ligands as CD28, avoiding T cell overactivation and hyperactivity. Similarly, programmed cell death protein 1 (PD-1; CD279) and its ligands PD-L1 (PD-1 ligand 1; CD274; B7-H1) and PD-L2 (CD273; B7-DC); or B7-H3 (CD276) and its ligandstriggering receptor expressed on myeloid cell (TREM)-like transcript 2 (TLT-2), toll-like receptor 2 (TLR2) or interleukin-20 receptor subunit alpha (IL20RA), that inhibit T cell effector functions and induce T cell apoptotic death [2,3,4,5,6]. Another IC is the cluster of differentiation 47 (CD47), which binds to signal regulatory protein alpha (SIRPα) of membrane macrophages to inhibit tumor cell phagocytosis [7].

The use of inhibitors against these ICs (immune checkpoint inhibitors (ICIs), also known as immune checkpoint blockade (ICB)) has shown clinical benefits in the treatment of different types of cancer. The inhibitors currently available are PD-1 inhibitors (i.e., Cemiplimab, Nivolumab and Pembrolizumab, and the recently approved Retifanlimab and Dostarlimab, approved in March and July 2023, respectively; www.fda.gov, accessed on 8 November 2023), and PD-L1 inhibitors (i.e., Atezolizumab, Avelumab and Durvalumab), and CTLA-4 inhibitors (Ipilimumab)), all approved by the U.S. Food and Drug Administration (FDA) [8]. However, treatment response depends on tumor type; good responses can be observed in immunogenic tumors, as seen in metastatic melanoma, which show a five-year overall survival (OS) rate of up to 52% when combining Nivolumab and Ipilimumab (compared to the five-year OS rate of about 35% for targeted therapy) [9]. Nonetheless, other tumors do not respond well to ICIs, and these inhibitors can even favor tumor progression, as has been observed in T cell leukemia–lymphoma after treatment with Nivolumab [10]. In addition, ICIs can activate a broad range of immune cells, resulting in immune-related adverse events (irAEs). Indeed, 43% of patients undergoing immunotherapy with ICI develop chronic or long-term adverse events, and up to 9% of ICI-treated patients may face severe or fatal consequences. The most common irAEs include skin toxicities, colitis, hepatitis, pneumonitis, nephritis, and endocrinopathies (i.e., thyroid abnormalities), although other less frequent effects such as cardiac disorders [8,11] may also appear.

This differential response to ICIs highlights the need to find novel biomarkers that can guide decision making to select the most personalized treatment for each patient. The FDA has approved different biomarkers for ICIs, such as PD-L1 expression in tumor cells, microsatellite instability (MSI), and Tumor Mutational Burden (TMB), referring to the totality of somatic mutations (single nucleotide polymorphisms (SNPs) and variations of copy number (CNVs)) per million bases. Other biomarkers are being studied, such as the tumor proportion score (TPS; evaluates expression of PD-L1 on tumor cells) [12], tumor immune dysfunction and exclusion (TIDE) signature (stratifies patients into high or low cytotoxic T lymphocyte count based on gene signature) [13], and Immunoscore (based on the density of total and cytotoxic tumor infiltrating T cell). However, these biomarkers present certain limitations of use for selecting the most appropriate individual treatment. Examples of this are that the effectiveness of ICIs may also benefit some patients with low PD-L1 expression [14] and the fact that TMB requires different cut-off values depending on tumor type [15]. Most of the immune cells infiltration models which are described above were based, originally, on the detection of immune cells in the invasive margin or inside the tumor. Unfortunately, in some cases, tumor biopsy is not always possible. Moreover, in some cancers, and particularly in tumor and its metastases counterpart, the correlation of these biomarkers with the response to ICI has not been completely elucidated [16,17]. In recent years, non-coding RNAs (ncRNAs) have attracted great attention for their ability to control diverse biological processes by targeting different molecular pathways. Among them, we can find: (1) small non-coding RNAs (sncRNAs), with less than 200 nucleotides, which include microRNAs (miRNAs), a class of highly conserved endogenous ncRNAs with approximately 22 nucleotides in length; and (2) long non-codingRNAs (lncRNAs) with a length above 200 nucleotides [1]. Various studies have reported the role of these ncRNAs as modulators of ICs molecules in the setting of human cancer [5,18,19,20] and even the relationship between them to regulate the response to immunotherapy, given that, for example, lncRNAs can act as competitive endogenous RNAs (ceRNAs) of miRNAs, establishing an lncRNA–miRNA–mRNA regulation axis [21].

Despite a growing number of studies on ICIs and ncRNAs generating valuable knowledge in various types of ncRNAs such as lncRNAs, most of the research has been found to be related with miRNAs. However, the role of miRNAs in ICIs response remains unclear [22]. In this systematic review, our aim was to compile a comprehensive list of miRNAs that have undergone experimental validation as regulators of response to ICIs. In addition, we selected papers presenting contrastable experimental evidence of how changes in miRNA expression impact the response to ICIs or vice versa. Finally, we performed a systematic review of the literature to gain deeper insight into the biological significance of miRNAs as potential predictors of ICI response. Additionally, our review highlights potential therapeutic targets that could enhance ICI therapeutic efficacy.

## 2. Materials and Methods

The entire search and selection process was carried out following Preferred Reporting Items for Systematic Reviewers and Meta-analysis (PRISMA) recommendations [23].

### 2.1. Search Strategy

The search was performed in March 2023. Articles were identified by two independent researchers (J.L.G.-G. and W.S.) searching the Web of Science (WoS), PubMed, Embase and Scopus using the following terms (immune checkpoint inhibitor OR immune checkpoint blockade) AND (miRNA OR microRNA) AND (cancer OR malignant neoplasm) and including all publications up to December 2022. Discrepancies in opinion between authors were arbitrated by a third author (S.M.-M.).

The full text of selected studies was consequently assessed for eligibility and included in an Excel file (Microsoft^®^ version 16.77) for further evaluation; duplicate articles were removed.

### 2.2. Screening and Eligibility Criteria

Based on the abstract, articles were selected for the next phase upon meeting the following criteria: (i) written in English, (ii) full text available, and (iii) original study (excluding reviews, systematic reviews, book chapters, meeting abstracts, etc.).

After full-text screening, we included only articles that fulfilled our main study objective of providing insights into either miRNA expression following ICI treatment or ICI impact assessment (direct confirmation of a better or worse response) after the upregulation or downregulation of miRNA levels.

As exclusion criteria, we used the following: studies that analyzed the relationship between miRNA and IC but did not provide results on the direct relationship of these miRNAs and the effectiveness of ICIs therapy; studies that used indirect ways to evaluate the efficacy of ICIs as immune cell infiltration (TIDE or others), TMB, MSI, expression levels of ICs (such as PD-L1) or other biomarkers; studies that did not experimentally evaluate the ICIS effect after modulating miRNAs levels or did not evaluate the difference in the expression of miRNAs before and after the treatment with ICIs; studies that performed only correlation between the levels of miRNAs and the effectiveness of the therapy without providing evidence that the alteration of the miRNAs levels altered the response to ICIS or that the use of the ICIs directly altered miRNA levels.

### 2.3. Data Extraction

The following information was retrieved from selected articles (if available): first author, year of publication, type of cancer, miRNA target gene, verification method of targeting, ICI evaluated, experimental model and effect on miRNA level or ICI response (effect on ICI response after experimentally modulating miRNAs levels, effect of ICIs therapy on miRNA levels, and irAEs observed due to the alteration of the miRNA levels produced by ICIs therapy). 

## 3. Results

The flowchart of the systematic review is depicted in Figure 1. The initial search yielded a total of 597 unique papers with 35% and 34% articles published in 2021 and 2022, respectively. Only 260 articles were original and therefore suitable for inclusion in this study.

A total of 22 studies published between 2017 and 2022 were finally included in our analysis. We retrieved 191 unique studies appearing with both ICI and ICB search terms, while 354 unique studies came from the ICI search alone, and 52 studies came from the search with ICB. ICI was the term used in the rest of our work as it yielded more search results than ICB.

Selected studies focused on ICIs, including anti PD-1 (seven studies); anti PD-L1 (eight studies); anti CTLA-4 (one study); combination of anti PD-1 and anti CTLA-4 (two studies); anti CD47 (two studies); anti CD276 (one study) and the combination of anti PD-1, anti PD-L1 and anti CTLA-4 (one study). 

Across the included studies, a total of 46 miRNAs were analyzed (let-7a, let-7b, let-7ev, miR-15b-5p, miR-16-5p, miR-18b, miR-20b-5p, miR-21, miR-22, miR-24, miR-25, miR-34a, miR-99a, miR-100, miR-128a, miR-143, miR-146a, miR-152, miR-155, miR-155-5p, miR-194, miR-214, miR-221, miR-324-5p, miR-335, miR-339, miR-339-5p, miR-340, miR-342-5p, miR-376c, miR-424, miR-424-5p, miR-429, miR-431, miR-433, miR-449a, miR-485-3p, miR-487a, miR-492, miR-582, miR-582-3p, miR-615-3p, miR-655, miR-708, miR-4649-3p, and miR-4759). These miRNAs were featured in only one study each except for miR-21 (two studies), miR-34a (two studies), miR-424 (two studies) and miR-155 (four studies).

The studies were conducted across 10 types of cancer: B-cell lymphoma, T-acute lymphoblastic leukemia, breast cancer, colorectal cancer, hepatocellular cancer, head and neck squamous cell carcinoma, lung cancer, melanoma, pancreatic carcinoma, and renal cell carcinoma.

The results from our analysis pinpointed 11 studies that elucidated the impact of miRNAs on ICI effect using the overexpression or downregulation of miRNAs (Table 1). Two further papers discussed miRNAs with altered expression patterns following ICI treatment (Table 2), while another three articles provided valuable insight into the role of miRNAs in irAEs (Table 3). 

Other articles were found examining correlations between miRNA levels and therapy response, but these were not included due to the fact that response to ICI was determined via biomarkers like IC expression, the TIDE algorithm, or the proportion of immune cells using CIBERSORT-type algorithms (a deconvolution method that characterizes the cell composition of the sample based on gene expression profiles) [24]. Other articles excluded were those which identified miRNAs targeting ICs but did not validate the impact of miRNA deregulation on ICI response. Lastly, studies analyzing miRNA levels only before or after treatment (common in biomarker identification studies) were omitted for not experimentally establishing the definitive link between changes in miRNA expression and ICI response. Appendix A provides a list of these excluded papers along with the reason for their exclusion.

**Table 1 ijms-25-01737-t001:** miRNAs that modulate response to immune checkpoint inhibitors (ICIs).

miRNA	Cancer	miRNA Target Gene	ICI	Experimental Model	Effect on ICI Response	Refs.
let-7a and let-7b	Head and neck squamous cell carcinoma	*TCF-4* *	anti CTLA-4	Overexpressing let-7a/b tumor cells inoculated into mice + anti CTLA-4	H	[25]
miR-15b-5p	Colorectal cancer	*PD-L1* *	anti PD-1	Overexpressing miR-15b-5p tumor cells inoculated into mice + anti PD-1.	H	[26]
miR-16-5p	Lung cancer		anti PD-L1	Tumor cell + overexpressing miR-16-5p exosomes + anti PD-L1	H	[27]
miR-20b-5p	Lung cancer	*PD-L1* *	anti PD-1	Tumor cells transfected with miRNA mimic + Pembrolizumab	H	[28]
	Breast cancer			Tumor cells transfected with miRNA mimic + Pembrolizumab	H	
miR-21	Oral squamous cell carcinoma	*PTEN*	anti PD-L1	Tumor cells inoculated into mice + miR-21 knockdown tumor-derived exosomes + anti-PD-L1	L	[29]
	Melanoma	*STAT1* *	anti PD-1	Tumor cells and knocked down miR-21 tumor-associated macrophages (TAM) subcutaneously injected in mice + anti PD-1	L	[30]
miR-128a	Laryngeal squamous cell carcinoma	*BMI1* *	anti PD-1	Overexpressing miR-128a tumor cells + Pembrolizumab	H	[31]
miR-155	Metastatic melanoma		anti PD-1 + anti PD-L1 + anti CTLA-4	Tumor cells inoculated into modified mice for knockout of miR-155 in CD4/8 T cells + anti PD-1, anti PD-L1 and anti CTLA-4	L	[32]
	Diffuse large B-cell lymphoma	*PD-L1* *	anti PD-L1	Overexpressing miR-155 tumor cells inoculated into mice + anti PD-L1	L	[33]
	Breast cancer	*SOCS1*	anti PD-L1	Overexpressing miR-155 tumor cells inoculated into mice + anti PD-L1	H	[34]
	Melanoma	*PD-L1* *	anti PD-L1	Overexpressing miR-155 tumor cells co-cultured with peripheral blood mononuclear cells + anti PD-L1	H	[22]
miR-340	Pancreatic carcinoma	*CD47* *	anti CD47	Overexpressing miR-340 tumor cells inoculated into mice + anti CD47	L	[35]
miR-424	Colorectal cancer	*CD28* and *CD80* *	anti PD-1 + anti CTLA-4	Tumor cells inoculated into miR-424 knocked mice + anti PD-1 and anti CTLA-4	L	[36]
				Mouse cecum orthotopic colorectal cancer + miR-424 knocked tumor cell-derived extracellular vesicles + anti PD-1 and anti CTLA-4	L	
	Hepatocellular carcinoma	*PD-L1*	anti PD-L1	Tumor cells inoculated into mice + nanobubbles carrying miR-424 mimic and anti PD-L1	H	[37]
miR-582	B-cell precursor acute lymphoblastic leukemia	*CD276* *	anti CD276	Overexpressing miR-582 tumor cells co-cultured with NK cells + anti CD276	H	[38]
miR-708	T-acute lymphoblastic leukemia	*CD47* *	anti CD47	Overexpressing miR-708 tumor cells + anti CD47.	H	[39]
miR-4759	Breast cancer	*PD-L1* *	anti PD-L1	Overexpressing miR-4759 tumor cells co-cultured with peripheral blood mononuclear cells + anti PD-L1	H	[40]

Abbreviations: Effect of miRNA levels on ICI response: H, high miRNA levels enhanced ICI efficacy; L, low miRNA levels enhanced ICI efficacy. miRNA target gene (*) indicates that it has been validated by luciferase reporter assay in the work.

**Table 2 ijms-25-01737-t002:** miRNAs modulated after response to immune checkpoint inhibitors (ICIs) in patients.

ICI	Experimental Model	miRNA	Experimental Effect on miRNA	Refs.
anti PD-1	miRNA analysis in peripheral lymphocytes from 21 good responders (complete response, partial response, or stable disease) with metastatic renal cell carcinoma before and after a 4-weeks period (2 cycles) of nivolumab administration	miR-99a, miR-708, miR-655, miR-582-3p, miR-492, miR-487a, miR-485-3p, miR-449a, miR-433, miR-431, miR-429, miR-376c, miR-342-5p, miR-340, miR-339-5p, miR-335, miR-324-5p, miR-25, miR-24, miR-22, miR-221, miR-214, miR-194, miR-18b, miR-152, miR-143, miR-100, miR-let-7ev	High levels of expression in peripheral lymphocytes after treatment compared to before treatment in good responders.	[41]
	miRNA analysis in peripheral lymphocytes from 17 good long-responders (complete response, partial response or stable disease and progression-free survival (PFS) > 18 months) with metastatic renal cell carcinoma before and after a 4-weeks period (2 cycles) of nivolumab administration	miR-22, miR-24, miR-99a, miR-194, miR-214, miR-335, miR-339, miR-708	High expression levels in peripheral lymphocytes after treatment compared to before treatment in good responders.	
anti CTLA-4 + anti PD-1	Plasma from stage IV melanoma non-responders (13 patients), partial response (4 patients) and complete response (5 patients) before and after Ipilimumab and Nivolumab/Pembrolizumab treatment	miR-4649-3p and miR-615-3p	Increased levels in post- vs. pre-treatment in non-responders. No changes post- vs. pre-treatment in patients with partial response. Decreased levels post- vs. pre-treatment in patients with complete response.	[42]

**Table 3 ijms-25-01737-t003:** List of immune-related adverse events (irAEs) related to miRNA and ICIs.

ICI	miRNA	Immune-Related Adverse Events (irAEs)	Refs.
anti PD-1	miR-34a	Anti PD-1 induced miR-34a-5p upregulation, modulating the miR-34a/Krüppel-like factor 4 (KLF4)-and miR-34a-5p/serine/threonine-protein phosphatase 1 regulatory subunit 10 (PNUTS) signaling pathway and inducing cardiac injury	[43,44]
anti PD-1 + anti CTLA-4	miR-146a	Downregulation of miR-146 increased the risk of developing irAE after ICI treatment. Toxicity was accompanied by increased infiltration of neutrophil and lymphocyte into the damaged organs.	[45]

### 3.1. miRNAs That Modulate Response to ICIs

According to Table 1 and Figure 2, increases in let-7a, let-7b, miR-15b-5p, miR-16-5p, miR-20b-5p, miR-128a, miR-582, miR-708, and miR-4759 levels are positively correlated with increased effectiveness of ICI therapy, while miR-21 and miR-340 presented a reduced response to ICIs in the analyzed models. miR-424 and miR-155 produced opposing outcomes according to the tumor type studied.

### 3.2. miRNAs Modulated after Response to ICIs

Other articles were found examining correlations between miRNA levels and therapy response, but these were not included due to the fact that response to ICI was determined via biomarkers like IC expression, the TIDE algorithm, or the proportion of immune cells using CIBERSORT-type algorithms (a deconvolution method that characterizes the cell composition of the sample based on gene expression profiles) [24]. Other articles excluded were those which identified miRNAs targeting ICs but did not validate the impact of miRNA deregulation on ICI response. Lastly, studies analyzing miRNA levels only before or after treatment (common in biomarker identification studies) were omitted for not experimentally establishing the definitive link between changes in miRNA expression and ICI response. Appendix A provides a list of these excluded papers along with the reason for their exclusion. 

Table 2 shows miRNAs modulated after response to ICIs in patients. Good responders (complete response, partial response, or stable disease) to ICIs saw increased miR-22, miR-24, miR-99a, miR-194, miR-214, miR-335, miR-339, and miR-708 levels, while the overexpression of miR-4649-3p and miR-615-3p correlated with no response to ICIs. 

### 3.3. miRNAs That Regulate Immune-Related Adverse Events (irAEs)

Finally, Table 3 shows that both upregulation of miR-34a and downregulation of miR-146A were associated with the appearance of irAEs.

### 3.4. miRNAs Related to ICIs Response

To analyze together all the miRNAs found, we made a Venn diagram (Figure 3), observing that only the miR-340 and the miR-708 had been studied in more than one category, “miRNAs that modulate response to ICIs” and “miRNAs modulated after response to ICIs in patients”. Among them, miR-708 was the only one with the same results in the different studies analyzed. While miR-340 was related to reduced response to ICIs in some studies, in others, its expression levels (in peripheral lymphocytes) after treatment with ICIs was associated with a good response.

## 4. Discussion

### 4.1. miRNAs That Modulate Response to ICIs 

From our literature review, we can conclude that several miRNAs directly or indirectly regulate PD-L1 expression (Table 1). The let-7 family (let-7a, let-7b, let-7c, let-7d and let-7e) is thought to mediate tumor suppression in cancer by inhibiting tumor cell proliferation, promoting cell death evasion or metastasis [46] but also alterations to immunity. Let-7 was significantly downregulated in tissue from head and neck squamous cell carcinoma patients compared to healthy tissue [25]. Let-7 downregulation has been observed in other types of cancer associated with reduced copy numbers, such as melanoma [47], with an upregulation of LIN28A/LIN28B, which is an RNA binding protein that inhibits Drosha or Dicer binding during let-7 biogenesis (breast cancer) [48], and with DNA hypermethylation (epithelial ovarian cancer) [49]. 

Let-7-5p-family miRNAs have been described as negative regulators of the pro-inflammatory and tumoricidal activity of “M1” macrophage tumor-associated macrophage (TAM) phenotypes (in contrast with the M2 phenotype with immunosuppressive and tumor-promoting activity), suppressing interferon (IFN)-γ-induced and immunostimulatory macrophage programming in cancer [50], and a luciferase reporter assay has demonstrated that T Cell Factor 4 (TCF-4) was a target of let-7a/b in head and neck squamous cell carcinoma [51]. TCF-4 forms a complex with β-catenin that activates N-glycosyltransferase STT3 transcription, maintaining PD-L1 glycosylation (critical for stability), binding to PD-1 and increasing PD-L1 expression [25]. Subcutaneous mice models were created with normal or let-7a overexpressing SCC7 cells, a mouse HNSCC line, and treated with anti CTLA-4. Let-7a/b overexpression in combination with anti CTLA-4 obtained the best results in terms of higher tumor lymphocyte infiltration and reduction in tumor volume [25]. Let-7 reduced the formation of the PD-1/PD-L1 complex, which allowed a role of tumor suppressor. Other authors have shown that treatment with a LIN28 inhibitor could re-establish let-7 biogenesis, reactivating T cell activity [52]. 

The miR-15 family (comprising miR-15a, miR-15b, miR-16, miR-195, miR-424, miR-497 and miR-503) targeted PD-1 [53] and presented an inverse correlation with PD-L1 in pleural mesothelioma [54]. In lung adenocarcinoma, miR-16-5p serum exosomes increased in a progression-dependent manner. Therefore, treating cells with high miR-16-5p-bearing exosomes in combination with anti PD-L1 increased tumor cell death. van Zandwijk and colleagues have proposed the use of TargomiRs, miR-16 mimic loaded epidermal growth factor receptor (EGFR)-targeted minicells, to improve ICI response, but this hypothesis has not yet been tested [55]. Cui et al. [56] defined the long non-coding RNA (lncRNA) C1RL-AS1/miR-16/Potassium Calcium-Activated Channel Subfamily N Member 4 axis (KCNN4; needed for efflux potassium during T cell activation [57] or NLRP3 inflammasome activation [58]). KCNN4 overexpression was related with low efficiency for ICIs in clear cell renal cell carcinoma. In another tumor model, with leukemia cells, the authors observed that miR-16-5p was transferred by tumor cells to T cells through microvesicles to induce T cell activation [59]. 

The overexpression of miR-15b-5p in tumor cells enhanced anti PD-1 sensitivity in colorectal cancer cell lines inoculated into mice [26]. In addition, miR-15b-5p was downregulated by nuclear respiratory factor 1 (NRF1), a transcription factor enriched in colorectal cancer, which is promoted by the elevated levels of interleukin IL-17A. Therefore, the authors proposed the co-treatment of anti-IL-17A (to reduce PD-L1 expression) and anti-PD-1 therapy to improve ICI efficacy [26,60].

miR-128a targeted Bmi1 (BMI1 proto-oncogene, polycomb ring finger), which is a major component of the polycomb group complex 1 (PRC1) and a factor that stimulates the degradation of p53, which is a gene found to be altered in a wide variety of tumors and which participates in response to immunotherapy [31]. Laryngeal cancer cells overexpressing miR-128a were treated with Pembrolizumab, resulting in greater cytotoxicity than in cells that do not overexpress miR-128a [31]. Mice with mouse pancreatic cancer cell line overexpressing miR-340 treated with an anti-CD47 developed a high presence of M1-macrophage and T cells, leading to increased tumor cell phagocytosis [35].

miR-4759 directly targeted and blocked PD-L1 expression, asvalidated by luciferase reporter assay. When human triple-negative breast cancer cells (MDA-MB-231 and BT-549) were transfected with miR-4759, treated with anti-PD-L1 and co-cultured with peripheral blood mononuclear cells, the in vitro cell killing assays showed enhanced tumor cytotoxicity [40].

Li et al. [38] observed that miR-582 overexpression increased NK cell-mediated cytotoxicity with B-cell precursor acute lymphoblastic leukemia cells upregulating the expression of B7-H3 (CD276), which is a newly discovered member of the B7 family that could promote T cell activation [61,62]. 

Finally, the blockade of CD47 with miR-708 produced phagocytosis and promoted apoptosis in a human CCRF-CEM leukemic T cell line and in human T-acute lymphoblastic leukemia Jurkat cells after transfection with miR-708 mimic. In another experiment, miR-708-overexpressed CCRF-CEM cells were incubated with macrophages derived from THP-1 (human leukemia monocytic cell line) and with anti CD47 antibody. Compared with treatment with anti CD47 only, miR-708 overexpression increased phagocytosis [39].

Jiang et al. observed that treatment with Pembrolizumab in NCI-H460 lung cancer cells and ZR-75-30 breast cancer cells produced a depletion of miR-20b-5p expression (measured by Real-Time Polymerase Chain Reaction, RT-PCR) and proteins PD-L1 and PD-1 (measured by Western blot). PD-L1 was confirmed as a target of miR-20b-5p in a luciferase assay. Moreover, the use of miR-20b-5p mimic improved the efficiency of Pembrolizumab therapy in cell models [28]. Phosphatase and tensin homolog (PTEN) is an antagonist of PI3K (phosphatidylinositol 3-kinase) and acts as a negative regulator of RTK/PI3K/Akt signaling, which regulates the expression of PD-L1 [63]. miR-20b targeted PTEN resulting in elevated levels of PD-L1 in advanced colorectal cancer [64]. Other miRNAs that target this pathway are miR-424 and miR-21 [65].

Zhao et al. observed that human colorectal cancer cells, secreted vesicles with high levels of miR-424 that blocked the CD28-CD80/86 costimulatory pathways in tumor-infiltrating T cells and dendritic cells, leading to immune checkpoint blockade resistance. Using a preclinical model of colorectal cancer (cecum-based orthotopic transplantation), the authors observed that intravenous injections of miR-424 led to the downregulation of tumor-secreted vesicles, which improved response compared with combined anti-PD-1 and anti-CTLA-4 therapy [36]. Liu et al. [37] described a new targeted therapy comprising an anti-PD-L1 conjugate and miR-424-loaded nanobubbles that was tested in subcutaneous mouse hepatocellular carcinoma. This combination decreased PD-L1 levels, increasing lymphocyte proliferation with a high tumor cell apoptosis rate [37]. Depending on the tumor type, miR-424 can act as a tumor suppressor (in hepatocellular carcinoma) or as an oncomiR (in colorectal cancer), and the therapeutic strategy would be to reverse miR-424 levels according to its role: that is, using miR-424 as a mimic if it acts as a tumor suppressor and blocking miR-424 synthesis when it acts as an oncomiR.

miR-21 is one of the most highly studied miRNAs. This miRNA is overexpressed in many types of cancer, and high levels are associated with poor prognosis [66]. miR-21 is an oncomiR, directly targeting several tumor suppressor genes, increasing the aggressiveness of tumor cells such as PTEN [65], programmed cell death 4 (PDCD4), a negative regulator of nuclear factor kappa B (NF-kB; an essential transcription factor involved in the regulation of immune checkpoint expression) [67,68] and signal transducer and activator of transcription 1 (STAT1), which is involved in M1-macrophage polarization. Therefore, the downregulation of miR-21 increases STAT1 and macrophage M1 polarization, increasing phagocytosis and anti-tumor immunity [30]. 

Using B16 murine melanoma cells and knocking down miR-21-TAMs subcutaneously injected in mice, melanoma tumors from miR-21-deficient mice showed a greater proportion of pro-inflammatory M1 TAMs vs. M2 TAMs. Therefore, treatment with anti-PD-1 antibodies offered superior anti-tumor activity compared to anti-PD-1 alone. [30].

Li et al. observed an enrichment of miR-21 levels in tumor-derived exosomes under conditions of hypoxia. According to the literature [64], the miR-21/PTEN/PD-L1 regulation axis enhanced the suppressive effect of immune cells in oral squamous cell carcinoma [29]. Thus, oral squamous cell carcinoma-bearing mice were treated with miR-21 knockdown tumor-derived exosomes with or without anti-PD-L1. The reduction in miR-21 potentiated the effect of anti PD-L1, reducing tumor volume [29]. 

Four reports about miR-155 and ICI response have been identified in our study. Overexpressed miR-155 produced an improved response to ICIs in breast cancer [34]. However, overexpression can also negatively affect response in B-cell lymphoma. In a xenograft model based on subcutaneous B-lymphoma cells transfected with miR-155 treated with anti-PD-L1, miR-155 upregulation induced an increase in PD-L1 expression and T cell apoptosis as well as inhibited tumor immunity [33]. 

In melanoma, the data were contradictory. On the one hand, the knockout of miR-155 in CD4+CD8+ T cells in modified mice enhanced the efficacy of immune checkpoint inhibitors (ICIs) when using B16 melanoma cells [32], On the other hand, in a human melanoma cell line mel1359 overexpressing miR-155 and co-cultured with peripheral blood mononuclear cells, there was an enhancement in the efficacy of ICIs [22]. This demonstrates that depending on the cellular context (such as the location of cancer cells, tumor growth phase, etc.), miRNA can either have a net oncogenic or net tumor-suppressive effect [69]. Despite this controversy, Huber et al. observed that exosomal miR-155 was associated with resistance to ICI treatment in melanoma patients [70], so increasing miR-155 levels seem an unwise strategy. Although, in contrast, phase II with MRG-106 (Cobomarsen), a miR-155 inhibitor, was terminated early due to the company’s decision [71].

Regarding the mechanism of action of miR-155, among various targets are PD-L1 [22,33,72], TCF4 [69], SOCS1 (Suppressors of Cytokine Signaling 1; a negative feedback regulator of cytokine and growth receptor factor signaling in t cells and promoter for M2 macrophage polarization) [73,74], and SHIP1 (SH2 domain-containing inositol 5’ phosphatase-1; a productor of inositol phosphate (direct product of PI3K and activator of Akt)) [75].

### 4.2. miRNAs Modulated after Response to ICIs

Metastatic renal cell carcinoma patients with good response (complete, partial response or stable disease) to Nivolumab treatment showed increased levels of 28 miRNAs in response to treatment. Among these, the overexpression of miR-22, miR-24, miR-99a, miR-194, miR-214, miR-335, miR-339, and miR-708 was associated with progression-free survival of over 18 months [41]. In contrast, miR-4649-3p and miR-615-3p overexpression was observed in stage IV melanoma patients who progressed after anti-CTLA-4 (Ipilimumab), anti-PD-1 (Nivolumab or Pembrolizumab), or the combination of Ipilimumab and Nivolumab [42]. miR-4649-3p has been identified as a tumor suppressor in triple-negative breast cancer by targeting PIP5K1C, an Akt activator [76], while miR-615 targets the NK group 2 (NKG2) family receptor or TNF-α from immune cells, leading to diminished cytotoxic activity against tumor cells [77].

miR-708 is a miRNA with the ability to modulate response to ICIs but is also modulated by ICIs. In both cases, its overexpression improved response to ICIs, although in contrast, it exhibited a tumor-suppressor role in both renal cancer cells [78] and T-acute lymphoblastic leukemia [79] unlike what is observed in other types of acute lymphoblastic leukemia (pre-B, pro-B, common). This miRNA can thus be considered a double-edged sword (oncomiR or tumor-suppressor role depending on tumor type).

### 4.3. miRNAs That Regulate Immune-Related Adverse Events (irAEs)

Xia et al. [44] published two interesting papers on the role of miRNAs in irAEs. One finding was an observed increase in miR-34a levels, which targeted and degraded Krüppel-like factor 4 (KLF4), an inhibitor of the M1 genetic program, and a significant increase in M1 macrophages and pro-inflammatory cytokines after treating mice with anti PD-1. This could be a cause of reduction in left ventricular ejection fraction and left ventricular fractional shortening (the fraction of the left ventricle shortens during a cardiac cycle). 

The same authors have demonstrated that macrophages secreted exosomes enriched with miR-43a after exposure to anti PD-1. These exosomes transferred miR-34a to cardiomyocytes, targeting the serine/threonine-protein phosphatase 1 regulatory subunit 10 (PNUTS) to induce cardiomyocyte senescence [43]. Other authors have associated miR-34a with Acute Respiratory Distress Syndrome [80], an inflammatory disorder resulting in rare fatal irAEs such as pneumonitis or respiratory failures [81], or other irAEs. Finally, further evidence of the need to maintain low miR-34a levels is demonstrated by the early closure of the Phase 1 study of MRX34, a liposomal mimic of miR-34a, in patients with advanced solid tumors, due to the death of several patients [82,83].

Genetic polymorphism rs2910164 in the *MIR146A* gene has been shown to reduce inflammamiR-146a expression and correlated with an increased risk of developing severe irAEs after ICI therapy [45,84]. In addition, miR-146a-5p induced myocardial inflammation and cardiomyocyte dysfunction in non-ICI therapy [85]. Furthermore, therapy with ICIs has been associated to increased NF-kB activity [60]. It is known that NF-kB pathway regulation is fine-tuned by miR-146, which in turn is a target of NF-kB. Interestingly, miR-146 downregulates tumor necrosis factor receptor-associated factor 6 (TRAF6) and IL-1 receptor-associated kinase 1 (IRAK1), leading to a signaling pathway involved in activating the regulatory subunit of IkB kinase (IKK), which in turn activates the expression of inflammatory genes regulated by NF-kB [86]. Therefore, a possible hypothesis would be that low miR-146 levels allow NF-kB overactivation, which causes oxidative stress and inflammation, triggering irAEs. Based on this evidence, miR-146a replacement therapy might be effective in patients with severe irAEs [45].

### 4.4. miRNAs Related to ICIs Response

miRNAs are molecules that change dynamically when the microenvironment changes, and they are sensitive to therapies and regulate a large number of cellular processes [87,88]. In addition, miRNAs are very stable in biological samples, which make them potential markers to be measured with several methods (i.e., small RNA-sequencing), which allow for identifying new associated miRNAs. Importantly, miRNAs can be also measured using more conventional techniques such as RT-PCR, making these molecules potential biomarkers to predict the response to therapy [89,90]. However, the fact that they are involved in many processes and miRNAs are upstream regulators of key transcriptional networks, acting as oncomiRs or tumor suppressors in different tumor types has open new avenues to use these miRNAs as potential markers in cancer. For example, miR-155 is a tumor suppressor of melanoma [91] and an oncomiR in breast cancer [92]. In fact, as we have observed for miR-155, miR-340 or miR-424, this signature may be associated with a good response to therapy with ICIs in some tumor types, while in other tumors, these miRNAs can be harmful. One of the limitations of this work is the fact that we have not found published studies investigating each miRNA in different tumor types; thus, it was not possible for us to identify whether the observed effect is general or specific to a particular tumor type. in this regard, it is necessary to evaluate the role of each miRNA in the response to ICIs for specific tumor types. Unfortunately, there is still much to investigate.

Regarding future cancer treatment, the development of mimics and inhibitors for miRNAs, as well as the fact that some are currently in clinical trials, suggests that the use of these experimental therapies will improve the response to ICIs. However, one limitation we have encountered in our research is that many studies utilizing miRNA mimics or inhibitors do not directly evaluate their impact on ICIs. Alternatively, they assess this effect indirectly using cellular models, often without considering the influence of the tumor microenvironment. Despite this, in our work, we were able to identify miR-708 as a miRNA that, when increasing its levels experimentally, provided a good response to therapy with ICIs in a model of T-acute lymphoblastic leukemia [39]. But also, in those patients with renal cell carcinoma where the levels of miR-708 increase in the blood after therapy with ICIs, the probability of survival increases [41]. Therefore, this work allowed for identifying a candidate to evaluate in future experiments using different tumor types and test whether the design of a miR-708 mimic will have the ability of improving immunotherapy.

## 5. Conclusions

The study of miRNAs and their clinical applications has a direct impact on cancer treatment development. miRNAs have demonstrated great potential for improving ICI treatment in preclinical studies. Moreover, miRNAs can be considered better biomarkers than other biomarkers such as PD-L1 expression or TMB for predicting response to ICI-based therapy. The reason is that miRNAs participate in several signaling pathways by regulating more processes than that produced by a simple accumulation of mutations (TMB) or expression of a single gene such as PD-L1. On the other hand, compared to those biomarkers that require a tumor sample, miRNAs can be detected in biological fluids such as blood or urine, facilitating non-invasive sample collection and reflecting the great heterogeneity that exists in a tumor [93]. This characteristic, together with the fact that the levels of miRNAs can be modulated throughout the treatment, as could be seen in Table 1, is what allows miRNAs to be postulated as excellent biomarkers to monitor therapy during treatment. However, to achieve this goal, more research is needed on the role of miRNAs in the response to ICI therapy, such as this systematic review. The expression of certain miRNAs influences the response to ICIs, and vice versa, so increasing our understanding of the role of specific miRNAs and allowing to elucidate the intricate mechanisms by which their antagonists act in response to ICIs. In this regard, further research would enable new druggable targets and biomarkers to be identified for delivering personalized cancer treatments. It has been shown that miRNAs can act as oncomiR as well as tumor suppressors. However, the behavior of miRNAs depends on tumor type. In fact, an increase in the levels of several miRNAs (let-7a, let-7b, miR-15b-5p, miR-16-5p, miR-20b- 5p, miR-128a, miR-155, miR-424, miR-582, and miR-4759) in tumor types where they act as suppressors, combined with a reduction in the levels of miR-21, miR-155, miR-340 or miR-424 in tumor types where they act as oncomiRs, could improve response to ICI therapy. Furthermore, given that ICI therapy has been shown to alter miRNAs levels in good responders (such as an increased expression of miR-22, miR-24, miR-99a, miR-194, miR-214, miR-335, miR-339, miR-708; or the reduction in miR-4649-3p and miR-615-3p), miRNAs are promising tools for monitoring therapy response, enabling researchers and clinicians to develop precision medicine aimed to improve ICI therapy. Other miRNAs to consider are those involved in the development of irAEs, among which attention must be paid to maintain low levels of miR-34a and high levels of miR-146a to reduce the risk of suffering these events.

Manipulating the expression of miRNAs by using miRNA mimics or miRNA inhibitors has been proposed as a promising therapeutic strategy for cancer treatment in recent years. Although its clinical use has not yet prospered, the use of targeted vehicles containing these therapies against miRNAs against ICs could improve results.

Finally, miR-708 has shown that by increasing its levels, it favors the response to the ICIs. Additionally, when miR-708 levels increase in patients treated with ICIs, it increases the probability of developing a good response, even in the long term, to ICIs therapy. Therefore, it is important to perform systematic revisions like this to identify potential biomarkers or even direct efforts towards the development of experimental therapies to modulate miRNAs levels such as miR-708.

## Figures and Tables

**Figure 1 ijms-25-01737-f001:**
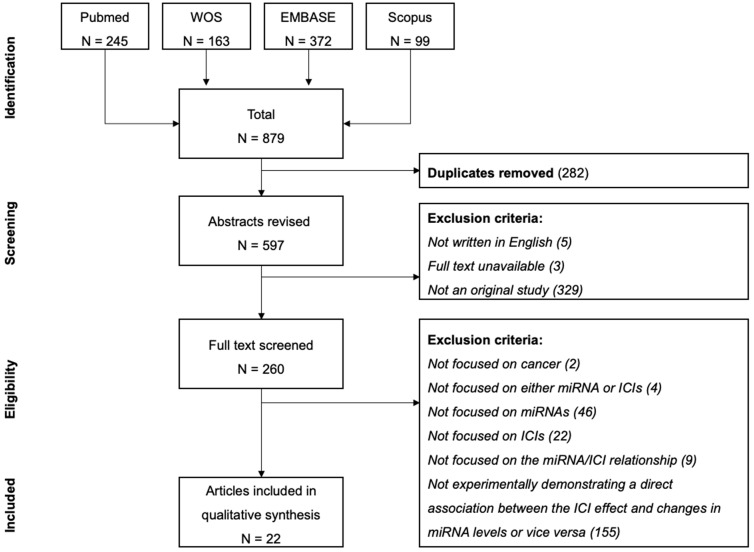
Workflow diagram of the literature review.

**Figure 2 ijms-25-01737-f002:**
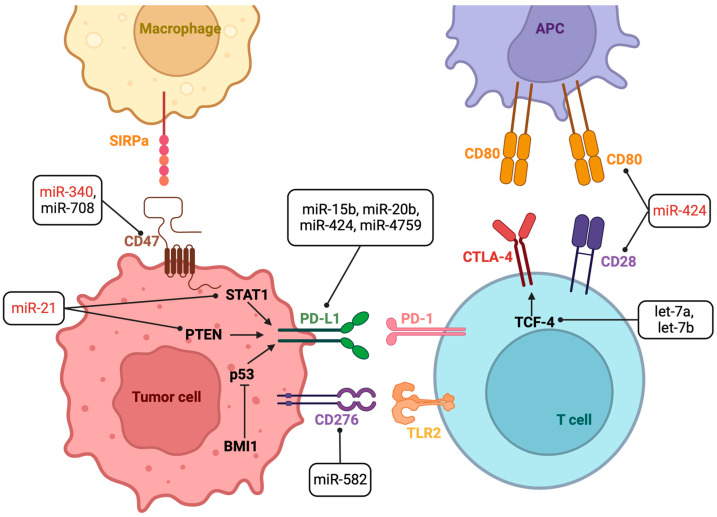
Scheme representing relevant miRNAs with the ability of modulating the effectiveness of immune checkpoint inhibitors (ICIs) by directly targeting ICs or other related modulators. Black: high miRNA levels enhanced ICI efficacy; Red: low miRNA levels enhanced ICI efficacy. APC: antigen-presenting cell; BMI1: BMI1 proto-oncogene, polycomb ring finger; CD47: cluster of differentiation 47; CD80: CD80 molecule (B7-1, CD28LG); CTLA-4: cytotoxic T lymphocyte-associated antigen-4 (CD152); p53: tumor protein p53; PD-1: programmed cell death protein 1 (CD279); PD-L1: programmed cell death protein 1 ligand (CD274; B7-H1); PTEN: phosphatase and tensin homolog; SIRPα: signal regulatory protein alpha; STAT1: signal transducer and activator of transcription 1; TCF-4: T Cell Factor 4; TLR2: toll-like receptor 2. Created with Bio.Render.com.

**Figure 3 ijms-25-01737-f003:**
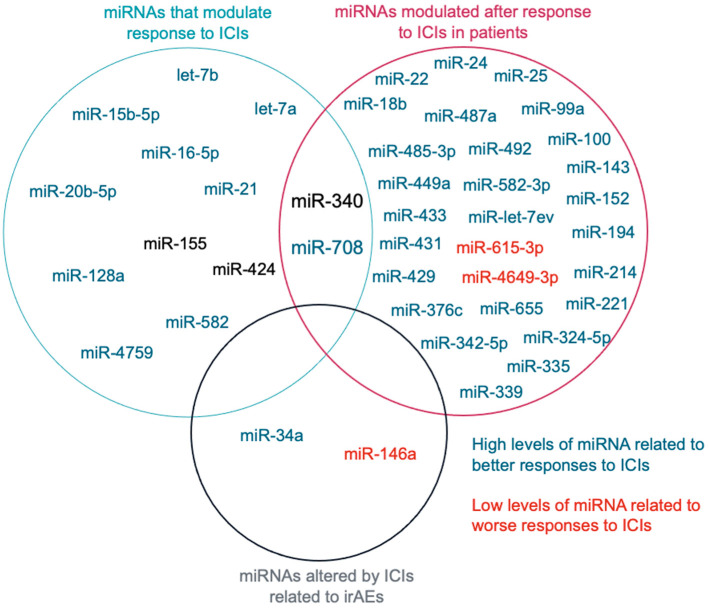
Venn diagram showing miRNAs that were identified as miRNAs able to modulate responses to ICI, miRNAs which were modulated after response to ICIs in patients, as well as miRNAs that can regulate irAEs.

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
