# Peer review of "miRNAs Related to Immune Checkpoint Inhibitor Response: A Systematic Review"

_ijms, 2024, doi:10.3390/ijms25031737_

Round 1

Reviewer 1 Report

Comments and Suggestions for Authors

As one of the non-coding RNAs, MicroRNAs play an important role in regulation of gene expression post-transcriptionally. In this review paper, the authors performed an extensive literature search and evaluated the clinical utility of miRNAs with a confirmed direct relationship with treatment response to immune checkpoint inhibitors. Overall, this manuscript is interesting and my specific comments are listed below.

1.    After carefully reading the full text, I don’t think it’s a good idea to emphasize “miRNAs as epigenetic regulators” in the title.

2.    As a review paper, it is highly recommended that the authors should combine their own related research work with other literatures to give some perspective comments.

3.    In the Introduction section, more backgrounds of miRNAs should be introduced, for example, it is one of the non-coding RNAs and other non-coding RNAs also play similar roles in the regulation of immune checkpoint inhibitors.

4.    It would be better if the authors can provide a figure to show us the mechanisms of the miRNAs in the Table 1 in various cancers.

Reviewer 2 Report

Comments and Suggestions for Authors

miRNAs as epigenetic regulators of immune checkpoint inhibitor response: a systematic review 

The premise of the review is to link miRNAs with immune checkpoint inhibitor response, which is an important aspect of drug resistance and therefore this systematic review has been proposed. 3 categories have been provided.

The abstract is generally ok.

The intro starts with some detail about T cell receptors and APCs and eventually, this is linked to CTLA-4, PD-1/ PD-L1 and CD47. As expected, ICIs then ensue and some examples have been provided. As an example, OS for melanoma has been provided, also an example of non-response.

The side effects of ICI then ensue and the authors describe irAEs and this is followed up by stats.

The next paragraph talks about biomarkers for ICIs which is how the authors have built momentum to get to the main premise of the study. Certain limitations of these biomarkers also follow and then miRNAs a molecular that may regulate ICs are provided.

The authors could outline why they think that these miRNAs might be a better biomarker candidate than the previously identified biomarkers.

Finally, the aims of the study are formulated.

One aspect that requires more emphasis is the importance of this study. What use does the finding of this study have and how can it direct future research.

Results:

The authors could indicate which of the authors (identified by their initials) performed any of the steps (such as screening of abstracts and articles and selection, mediation of any disputes).

The flow chart is useful.

The authors point out the number of studies identified for each IC. 46 in total.

The authors try to be inclusive by including many cancer types. This, however, raises the question of specificity. Would the conclusions made be relevant to all these cancers?

11 studies specifically investigate the link between miRNAs and ICIs. 2 about the regulation of miRNA by ICIs and then irAEs.

The authors should look through the manuscript and make sure the aims they give match, are they looking for links between ICIs and miRNAs or ICs and miRNAs, since I have seen both forms being used and it makes a difference.

In Table 1, the miRNA positively and negatively correlated with ICI are summarised. The experimental model is also incorporated.

In Table 2, those miRNAs regulated by ICIs then appear. This table affects miRNA as well compared to Table 1. Can Table 1 have something to that effect (such as mechanism etc)?

And finally, table 3, that links miRNAs with irAEs.

The discussion then talks about each of the findings, for example, miRNAs that regulate any of the ICIs which is good. The discussion is also structured which makes it easier to follow.

The authors could add a summary figure to show the various miRNAs found and the category they were allocated.

One aspect that can be added to the discussion is the significance of the work, its limitations and future directions. 

Methods:

The authors could also release their extended search strings.

Also, they could ensure if any other databases were left out (WoS, PUBMED, MEDLINE, Scopus and Embase) but were any other databases searched.

The authors should clearly state the exclusion criteria as well. Please outline who did what task.

They state the type of data they extracted.

Overall, this is a short methods section.

This is followed by some concluding remarks.

Comments on the Quality of English Language

Some editing required

Round 2

Reviewer 2 Report

Comments and Suggestions for Authors

The authors have addressed my comments